# Noncontact Optical Measurement of Aqueous Humor Glucose Levels and Correlation with Serum Glucose Levels in Rabbit

**DOI:** 10.3390/bios11100387

**Published:** 2021-10-13

**Authors:** Yih-Shiou Hwang, Eugene Yu-Chuan Kang, Chia-Rui Shen, Wei-Hsin Hong, Wei-Chi Wu

**Affiliations:** 1Linkou Medical Center, Department of Ophthalmology, Chang Gung Memorial Hospital, Taoyuan 333, Taiwan; sub170324@yahoo.com (E.Y.-C.K.); sub29994@yahoo.com.tw (W.-C.W.); 2College of Medicine, Chang Gung University, Taoyuan 333, Taiwan; 3Department of Ophthalmology, Chang Gung Memorial Hospital, Xiamen 361000, China; 4Department of Ophthalmology, Jen-Ai Hospital Dali Branch, Taichung 412, Taiwan; 5Graduate Institute of Clinical Medical Sciences, Chang Gung University, Taoyuan 333, Taiwan; 6Department of Medical Biotechnology and Laboratory Science, College of Medicine, Chang Gung University, Taoyuan 333, Taiwan; sub171855@gmail.com (C.-R.S.); sub170561@gmail.com (W.-H.H.)

**Keywords:** noncontact, glucose, measurement, aqueous humor, serum, optical

## Abstract

The noninvasive measurement of serum glucose levels has been investigated for the monitoring of blood sugar control in diabetes. In our study, we aimed to develop a novel noncontact glucometer (NCGM) utilizing an optical approach to measure the intraocular aqueous humor glucose levels in the anterior chamber of rabbit eyes. The NCGM consists of a hybrid optical system that simultaneously measures near-infrared absorption and the polarized rotatory distribution of glucose molecules in the aqueous humor. In vitro optical measurements demonstrated that NCGM measurements had high precision and repeatability for different glucose levels, including 50 mg/dL (14.36%), 100 mg/dL (−4.05%), 200 mg/dL (−5.99%), 300 mg/dL (4.86%), 400 mg/dL (−2.84%), 500 mg/dL (−0.11%), and 600 mg/dL (4.48%). In the rabbit experiments, we found a high correlation between aqueous glucose levels and serum glucose levels, with a mean difference of 8 mg/dL. According to the testing results, the in vivo NCGM measurement of aqueous humor glucose levels also displayed a high correlation with serum glucose levels, with a mean difference of 29.2 mg/dL. In conclusion, aqueous humor glucose levels were accurately measured using the NCGM, and the results correlated with serum glucose levels.

## 1. Introduction

An estimated 1 in 11 adults aged between 20 and 79 years was living with diabetes mellitus (DM) in 2015 [1]. In 2017, the global incidence of DM was estimated at 451 million people, and this number is expected to increase to 693 million by 2045 [2]. The increase in the incidence of DM has been ascribed to an aging global population, continued economic development and urbanization, unhealthy eating habits, and an increase in the proportion of people with inactive lifestyles [1]. The economic impacts associated with DM are considerable; in 2017, the economic burden for the USA associated with diagnosed and undiagnosed DM, gestational DM, and prediabetes was estimated at $404 billion. The financial burden of DM highlights the importance of improving disease detection, prevention, and treatment [3].

People with DM frequently experience hypoglycemia and hyperglycemia, especially patients treated with insulin [4]. Therefore, an essential component of DM management involves the maintenance of blood glucose levels within the normal range, between 80 and 150 mg/dL, which requires frequent glucose monitoring [5]. Self-monitoring blood glucose (SMBG) devices and continuous glucose monitoring (CGM) systems are the most commonly used devices for this purpose, with CGM systems becoming increasingly popular in high-income countries. SMBG devices remain widespread, particularly among the 80% of people with DM who reside in low- to middle-income countries [4]. The preferred method for point-of-care and personal blood glucose monitoring is an electrochemical method that uses glucose oxidase and represents an affordable and straightforward method that is easy to perform. This method is highly accurate and specific and has limited cross-reactivity; however, the technique is invasive, requiring a small blood sample. Repeated blood sampling can be painful, cause discomfort, raise infection risk, and lead to tissue damage. Patients often become non-compliant due to blood sampling requirements and the ongoing cost of testing strips [6]. Therefore, a concerted effort has focused on the development of less invasive techniques for glucose monitoring.

Two alternative types of glucose monitoring technologies are currently being pursued: minimally-invasive (MI) and noninvasive (NI). MI techniques require bodily fluids, such as tears or interstitial fluid, to monitor glucose using electrochemical means, whereas NI techniques can monitor glucose via various forms of radiation. NI technologies based on optical techniques using visible and infrared light are the most extensively studied. Several commercial NI devices have been developed for glucose monitoring, and most target the arm or the ear; however, no currently available device makes use of optical polarimetry [6].

Optical polarimetry is a technique that allows for the measurement of glucose in the aqueous humor in the anterior chamber of the eye. Glucose is a chiral molecule able to rotate the polarization plane of light. A laser beam (with a frequency between 400 and 750 nm) is polarized and directed into the eye, and the degree of rotation is proportional to the glucose concentration. The advantages of this technique include a very high resolution and the ability to miniaturize the optical components [6]. Polarimeters can now accurately measure changes in glucose levels smaller than 10 mg/dL, falling well within the range necessary to monitor physiological glucose levels [7]. However, this method is sensitive to changes in temperature and motion; the readings can be affected by optically active compounds; and a lag time exists between blood glucose levels and glucose levels in the aqueous humor [6]. Several prototypes have been developed that are able to measure glucose based on the rotation of polarized light in the eye [7,8].

In our study, we aimed to develop a convenient, noncontact machine for measuring glucose levels. The prototype device, referred to as a noncontact glucometer (NCGM), uses the simultaneous measurement of near-infrared absorption and the polarized rotatory distribution of glucose molecules in the aqueous humor of the eye. The aims of this study were as follows: (1) to perform an in vitro study to examine the precision and accuracy of optical readings obtained by the NCGM for different glucose concentrations; (2) to perform an in vivo study to investigate the precision and accuracy of NCGM optical readings for monitoring glucose levels in the aqueous humor of rabbit eyes; and (3) to examine the correlations between NCGM optical readings, rabbit aqueous glucose levels, and rabbit serum glucose levels.

## 2. Materials and Methods

### 2.1. NCGM

A diagram of the electro-optic system designed for use in this study is depicted in Figure 1. The device consisted of a light source, two beam splitters, three photodetectors, and a processing unit for analysis. The light source, equipped with a laser diode, produced a beam of optical polarized light with a wavelength of 1650 nm and a power of 5 mW. After the light was emitted from the light source, it passed through the beam-splitter, equipped with a focusing function, and was divided into two light beams and directed to the eye to be tested and a photodiode (for optical feedback control). The light reflected from the anterior chamber of the test eye was also divided into two light beams by the beam-splitter and directed to the photodiode for optical feedback control and a further beam splitter. This beam splitter then directed the light to two photodiode detectors that measured the optical rotatory distribution (ORD) and absorption energy of the reflected light. After receiving information from the reflected light, the processing unit analyzes the ORD and absorption energy to calculate differences in the ORD and absorption energy of the collected light compared with the emitted light. Information regarding the glucose level is obtained after the processing unit has completed its analysis. For safety, we applied the international standards for medical laser products published by the International Electrotechnical Commission, and the machine passed the corresponding tests.

### 2.2. Repositioning Error Correction of NCGM

The NCGM device is equipped with an alignment system for repositioning error correction, which ensures that the detection of the intraocular reflected optical signal is received from the same spot. The system generates light spots on the eye’s surface, an alignment light spot, and a reference light spot. The alignment method captures and analyzes an image comprising the reference light spot, the alignment light spot, and the eye. We can then set a starting position point for these light spots and the eye image to compare it with information from other positions. When the alignment is performed, the positions of the current light spots are compared with the starting position. If the positioning is within an acceptable range, data for glucose analysis are recorded. Using the repositioning error correction technique, the range of the allowable error was ± 3 pixels within a captured image of 480 × 640 pixels (Figure 2).

### 2.3. In Vitro Study and Repeatability Test

Before applying the NCGM to animal experiments, we used a cuvette to perform an in vitro study and repeatability test. Glucose powder was diluted to specific concentrations in a balanced salt solution (Alcon Laboratories Inc., Fort Worth, Texas) and poured into the cuvette. After the machine was calibrated using distilled water, the absorption and polarization of seven concentrations of glucose solution (50, 100, 200, 300, 400, 500, and 600 mg/dL) were tested. The NCGM measured the optical signal of the glucose solutions in the cuvette. Each solution was measured for 20 minutes with a recording frequency of 10 Hz (10 times per second) to produce 12,000 measurements and consecutive sets of 1000 measurements were averaged and plotted.

### 2.4. Animal Experiments

All animal experiments were performed in strict adherence to the Association for Research in Vision and Ophthalmology Statement for the Use of Animals in Ophthalmic and Vision Research and in accordance with the National Institutes of Health guidelines. Animal protocols were reviewed and approved by the Chang Gung University Animal Research Committee (No. 201301153A0D001). Four rabbits were used in this study.

The protocol used for animal experiments, including blood and aqueous humor sampling, is shown in Table 1. The rabbits were fasted for at least 8 hours before the NCGM measurement and until the aqueous sampling ended. Before the experiment, preparation steps including machine warm-up (less than 5 minutes); collecting rabbit’s characteristics; inhalant anesthesia with Forane (isoflurane, Healthcare Corporation, Deerfield, IL, USA) according to the rabbit’s weight; monitoring; positioning; and shaving of the periocular area were performed. After all preparations were made and anesthesia was administered, NCGM measurements were taken every 10 mins, continuously, for a total of 6 times from one eye. An intravenous glucose challenge consisting of 1.5 mL of Vitagen 50% glucose water (Taiwan Biotech Co., Taoyuan, Taiwan) was infused at 20 min. The aqueous humor was extracted by tapping the anterior chamber of both eyes. One eye (the non-NCGM–measured eye) was sampled before the intravenous glucose challenge, whereas the other eye (the NCGM-measured eye) was sampled 40 minutes after the challenge. During NCGM measurements, 1 mL-blood samples were collected through ear phlebotomy and sent for glucose tests. Standard preoperative preparation for the aqueous humor extraction included topical anesthesia, 5% povidone–iodine, and the use of a sterile lid speculum. A 30-gauge 1-inch needle was used for anterior chamber paracentesis, and 0.1 mL of the aqueous humor was collected. The specimen was transferred to small Eppendorf tubes for transportation and remained frozen (−20 °C) until glucose measurements were performed.

### 2.5. Methods Used for Glucose Measurement

For improved accuracy in the glucose measurement of the collected samples, we used the hexokinase/glucose-6-phosphate dehydrogenase method with deproteinization, which has been accepted as the standard reference method for glucose measurement (Fisher Diagnostics, Middletown, VA, USA) [9].

### 2.6. Statistical Analysis

The agreement between two methods (i.e., aqueous glucose vs. serum glucose) was assessed using two methods. First, a Bland–Altman plot was used to plot the mean glucose levels from two methods against the difference between the two methods. The agreement between the two methods was considered acceptable when the individual differences between the two methods did not exceed the tolerated boundaries, usually the 95% confidence intervals (CI) of the mean difference between the two methods. Second, the intra-class correlation (ICC) of the consistency between the two methods was calculated. The ICC value was obtained under a two-way mixed-methods model in which the observations were set as a random effect, and the raters (the two methods) were set as a fixed effect. A significant value of ICC > 0.7 was considered to indicate good agreement between the two methods. A two-sided P-value < 0.05 was considered significant. The Bland–Altman plot and ICC were conducted using MedCalc Statistical Software version 13.1.2.0 (MedCalc Software by Ostend, Belgium).

## 3. Results

### 3.1. In Vitro Study and Repeatability Test

Each solution was measured for 20 min, with a recording frequency of 10 Hz (10 times per second, Figure 3). The standard errors for the predictions are presented in brackets next to each glucose solutions for the following concentrations: 50 mg/dL (14.36%), 100 mg/dL (−4.05%), 200 mg/dL (−5.99%), 300 mg/dL (4.86%), 400 mg/dL (−2.84%), 500 mg/dL (−0.11%), and 600 mg/dL (4.48%). The results fit the standard for in vitro glucose measurements, established in ISO 15197:2013, which requests a variation of the glucose level within 15% of a target value ≥ 100 mg/dL or 15 mg/dL for a target value < 100 mg/dL.

### 3.2. Correlation between Aqueous and Serum Glucose Levels

Before and after the intravenous glucose challenge, the glucose levels measured in the aqueous humor were compared with the nearest serum glucose measurements. The agreement between the aqueous glucose and serum glucose levels is illustrated in Figure 4 (raw data in Appendix A). The mean difference was 8 mg/dL among the 8 observations across 4 rabbits. These observations did not exceed the upper and lower boundaries of the 95% CI. The ICC was 0.82, with a 95% CI between 0.37 and 0.96 (P = 0.004).

### 3.3. Glucose Measurements Using NCGM for Rabbit Eyes

The agreement between the NCGM measurements and blood glucose levels at different testing times is shown in Figure 5. The mean difference was 29.2 mg/dL among 28 observations obtained in 4 rabbits. Only 2 observations exceeded the lower boundary of the 95% CI. The ICC was 0.55, with the 95% CI between 0.15 and 0.78 (*P* < 0.001).

## 4. Discussion

In this study, a new device, the NCGM, was developed to measure the hybrid profile of 2 types of optical signals for the detection of glucose levels in the aqueous humor. The photodetectors detected 2 combined signals from the light reflected from the anterior chamber of the eye to calculate the glucose levels in the aqueous humor. Using this device, we were able to demonstrate strong correlations among the following three parameters: the reading of the device, the measured aqueous humor glucose level, and the measured serum glucose level. The use of the aqueous humor of the eye to measure glucose in the optical system has previously been reported [7,8]. In both prior studies, the researchers used a contact-based coupling method, and the authors stated that a commercially viable technique would require the development of a noncontact method [7]. To our knowledge, this hybrid optical system represents the first noncontact hybrid optical glucose measurement system to demonstrate a high correlation with serum glucose levels in rabbits.

Modern clinical practice emphasizes the development of techniques able to detect early changes in disease activity when new treatments tend to be extremely effective and irreversible damage can be prevented or managed. Because glucose levels are recognized as among the most vital prognostic markers in patients with DM, developing methods for easy and convenient blood sugar monitoring with high precision is especially critical for controlling DM-associated pathology. The current “finger stick” method of measuring blood glucose is a multistep process that is particularly difficult for younger and older patients, those with impaired vision, and those with impaired hand control. The user pierces the skin with a lancet, squeezes a drop of blood onto a test strip, and inserts the strip into a meter from which the glucose value is read. This process is painful, carries the risk of infection, can damage nerves, and is unpopular among young children and teenagers. Therefore, various NI technologies have been introduced during the past 2 decades. Tissue fluid samples other than blood, such as sweat and saliva, have been measured using subcutaneous, dermal, epidermal, and combined dermal and epidermal sensing approaches, with biochemical and optical sensing methods representing the most commonly used approaches. The biochemical method detects glucose by applying glucose-sensitive reagents to the body sample and measuring the resulting changes in the biochemical reactions [10]. Single-use strips remain necessary for many developed commercial glucose meters. Glucose-sensing electrical circuits built into contact lenses have been reported [11]; however, the glucose levels in tears may reflect air humidity, which might interfere with precision.

The optical method measures the intrinsic molecular properties of glucose in body samples other than blood, such as the absorption of near-infrared or mid-infrared radiation [12,13], polarimetry [14], Raman spectroscopy [15], photo-acoustic absorption [16], light scattering [17], optical coherence tomography [18], confocal reflectometry [19], fluorescence spectroscopy [20], and contact lenses [21,22]. The main concerns associated with glucose-measuring optical technologies include the difficulties achieving sufficient accuracy at an affordable cost for home-care users without the use of blood to perform the measurement. Most optical technologies for glucose measurement use a single method to track a physiological phenomenon correlated with blood glucose levels; however, several other biological constituents can interfere with such signals. Thus, these techniques generally demonstrate low specificity and sensitivity. A multi-sensing approach can be used to reduce the errors associated with each individual sensing method, increasing the accuracy by increasing signals relative to the low accuracy achieved by a single sensing technology.

In the 1980s, Rabinovitch et al. performed a pioneering study describing the NI determination of glucose concentration in the aqueous humor of the eye by measuring the rotation angle of polarized light caused by the optical activity of glucose [23]. Since then, the aqueous humor of the eye has become a common site for the development of NI optical approaches to investigate glucose levels. The aqueous humor is a transparent and gelatinous fluid in the anterior and posterior chambers of the eye, which are located between the cornea, lens, and vitreous. Because the cornea is transparent, polarized light can be passed through it to measure the rotation angle induced by glucose present in the anterior chamber, which is approximately 3 mm in depth. However, these measurements can be complicated by the birefringent property of the cornea, which exhibits multiple refractions of polarized light and scatters the light into 2 paths [8]. The aqueous humor represents a type of blood ultrafiltrate secreted from the ciliary epithelium to support the lens. The aqueous humor constitutes an essential component of the optical system and is involved in physiologically critical processes for eye function. The electrolyte concentrations of Na+, K+, and Cl− in the plasma and aqueous humor are similar. The protein concentration in the aqueous humor is approximately 200 times lower than that of plasma. The ascorbate concentration of the aqueous humor is 20–50 times higher than that of plasma. The glucose, lactate, and urea concentrations in the aqueous humor are approximately 80% of those in plasma. Because lactate, ascorbate, and urea are also optically active molecules, these biological constituents interfere with the measurement of glucose when using the polarization method. The principal issues encountered when measuring glucose in the aqueous humor using optical approaches include corneal birefringence, low signal-to-noise ratio, motion artifacts, and the potential time lag between blood and aqueous humor concentrations during rapid changes in glucose levels.

The optical measurement of glucose levels in the aqueous humor requires calibration, which we were able to perform using our machine. We applied a single-person calibration concept by using simultaneous blood glucose measurements. Therefore, at least one serum glucose test is required at the time of the first NCGM measurement. We did not use the aqueous glucose level as a calibration reference because patients will not be able to obtain aqueous humor glucose levels at home if they require recalibration. A time lag has previously been reported between the blood glucose values and the aqueous humor glucose levels, although this lag was reported as shorter than 5 minutes in rabbit eyes [24]. In a different study, the aqueous humor glucose levels in the rabbit’s eye lagged blood glucose values by 30 minutes [25]. In our study, this bias could be excluded because we fasted the animals for at least 8 hours until all sample collections were finished.

Some issues remain that require further investigation. First, the aqueous humor contains several molecules, electrolytes, and proteins [26], and the potential for interference from these contents should be considered. We examined the polarization and absorption of other components and observed that only vitamin C interfered with the polarization recording (data not shown). If a participant is taking vitamin C regularly, that participant should be excluded from the use of this device. Second, polarization recording requires a body part with low scattering, such as the cornea, for appropriate calibration. The eye offers an advantage over the skin for NI measurements of glucose due to the absence of scattering [27]. Corneal rotation, corneal birefringence, and eye motion artifacts, however, remain potential sources of error during polarimetric ocular measurements [27]. For this purpose, NCGM recordings were collected rapidly, within seconds, to avoid eye motion artifacts. However, the accuracy of this measurement method in cases with corneal pathologies requires further study.

## 5. Conclusions

Our NI, noncontact hybrid optical glucose monitor accurately measured aqueous glucose levels in rabbit eyes. The monitor reading, aqueous glucose levels, and serum glucose levels demonstrated high correlations. This novel technology may complement conventional blood tests for monitoring glucose levels in patients with DM.

## 6. Patents

Patents issued to the device by the United States Patent and Trademark Office included (1) Taiwan Biophotonic Corporation. Apparatus for non-invasive blood glucose monitoring. US9833175B2, 05 Dec. 2017. (2) Taiwan Biophotonic Corporation. Method for non-invasive blood glucose monitoring and method for analysing biological molecule. US9743864B2, 29 Aug. 2017; (3) Taiwan Biophotonic Corporation. Device and method for alignment. US9696254B2, 04 Jul. 2017. (4) Taiwan Biophotonic Corporation. Optical measurement device and optical measurement method. US10048197B2, 14 Aug. 2018. 

## Figures and Tables

**Figure 1 biosensors-11-00387-f001:**
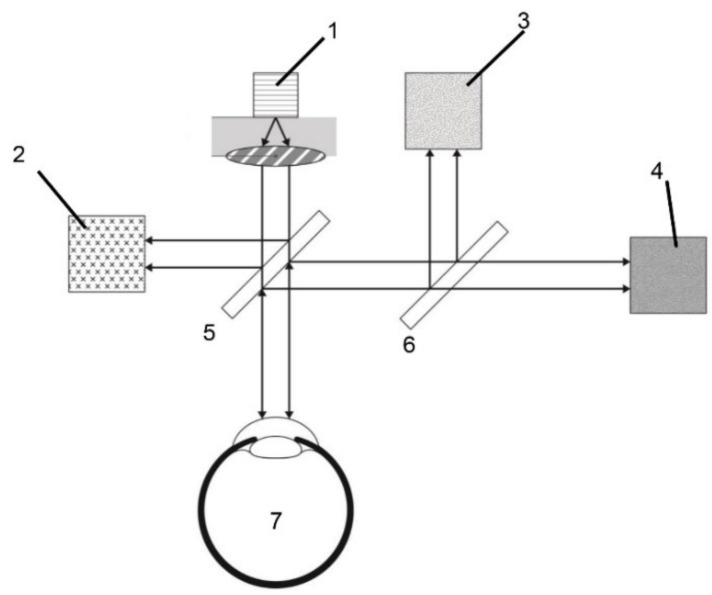
Schematic illustration of the noncontact glucometer system. (1) Laser diode, (2) photodiode for optical feedback control, (3) photodiode for detected absorption of light, (4) photodiode for detected polarization of light, (5) and (6) beam splitter, and (7) testing eye.

**Figure 2 biosensors-11-00387-f002:**
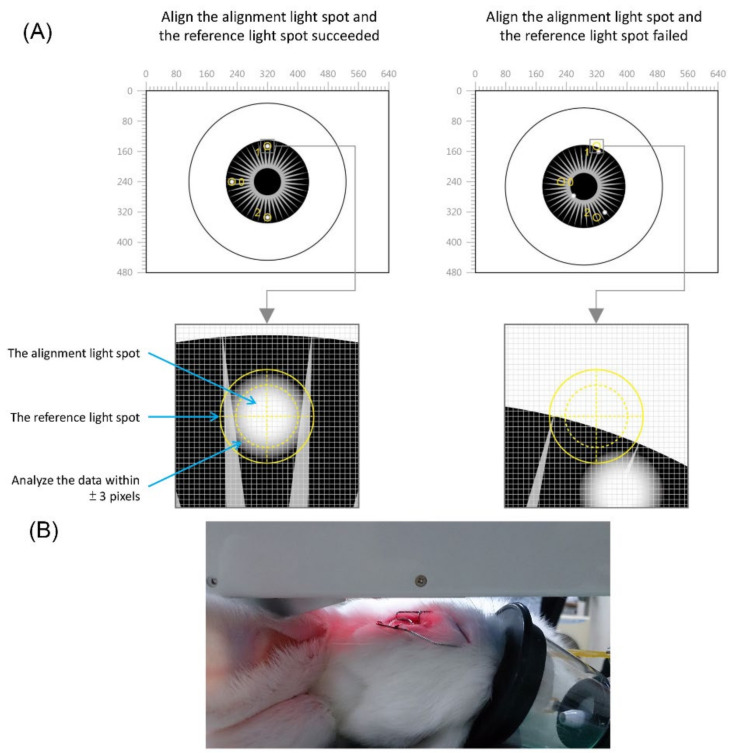
(**A**) Illustration of the alignment system for repositioning error correction in the noncontact glucometer (NCGM) device. (**B**) the NCGM device was placed in front of the rabbit’s eye.

**Figure 3 biosensors-11-00387-f003:**
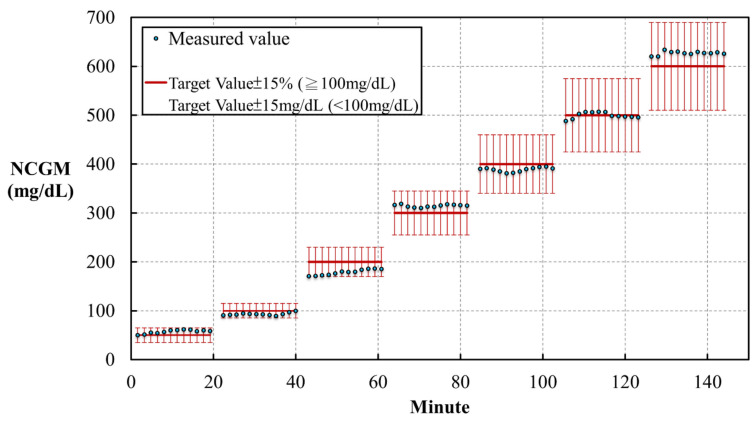
The results of repeatability tests in the in vitro study using a noncontact glucometer (NCGM) to measure different glucose concentrations.

**Figure 4 biosensors-11-00387-f004:**
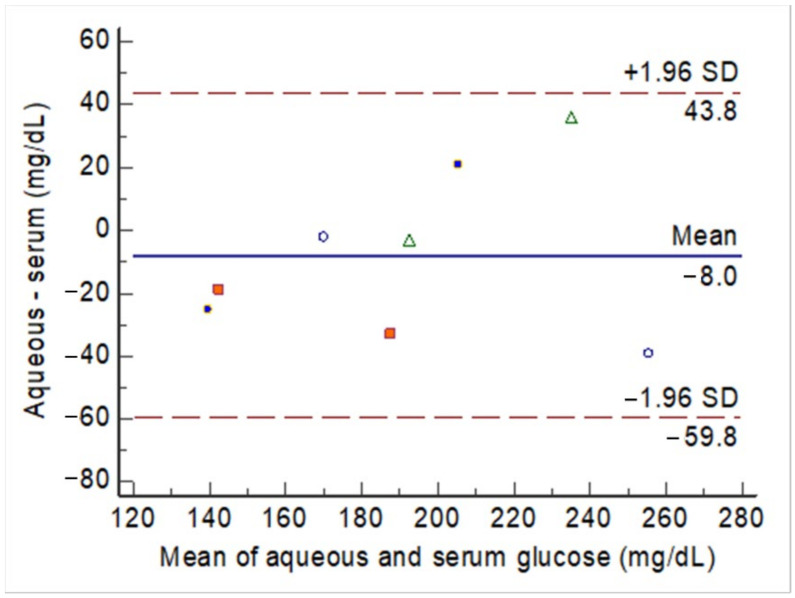
The Bland–Altman plot for the agreement between aqueous glucose levels and serum glucose levels before (20 min) and after (60 min) the intravenous glucose challenge. Each rabbit had two observations, which are indicated by marks of the same shape and color.

**Figure 5 biosensors-11-00387-f005:**
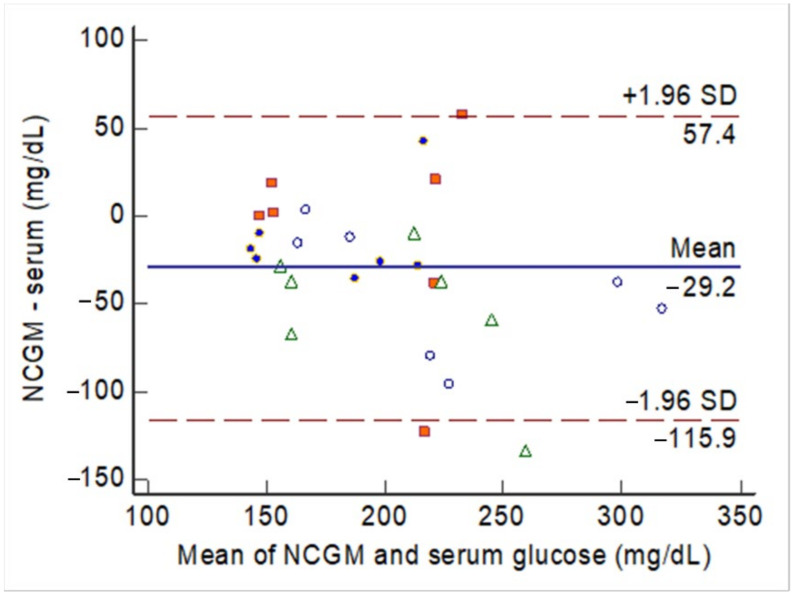
The Bland–Altman plot for the agreement between glucose levels measured by the noncontact glucometer (NCGM) and serum glucose levels during the animal experiments. Each rabbit had multiple observations during the experiments, with each animal indicated by marks of the same shape and color of the mark.

**Table 1 biosensors-11-00387-t001:** The protocol used for the animal experiments in our study.

	Preparations	
	Noncontact glucometer (NCGM) warm-up	
	Measuring rabbit’s characteristics	
	Anesthesia for the rabbit	
	Vital sign monitoring, positioning, and shaving of the rabbit	
**Time (mins)**	**Animal experiment process (4 rabbits)**	
0	Collecting blood samples from ear phlebotomy 3 times (baseline and every 10 mins)	Measuring aqueous glucose level by NCGM 3 times (baseline and every 10 mins)
10		
20		
	Aqueous tapping over the non-NCGM-measured eye	
	Intravenous glucose boost with 1.5 mL of 50% glucose	
30	Collecting blood samples from ear phlebotomy 4 times (every 10 mins)	Measuring aqueous glucose by NCGM 4 times (every 10 mins)
40		
50		
60		
	Aqueous tapping over the NCGM-measured eye	
	**Experiment end**	

## Data Availability

The data presented in this study are available on request. The data are not publicly available due to the data security policy of Chang Gung Memorial Hospital.

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
