# Peer review of "Noncontact Optical Measurement of Aqueous Humor Glucose Levels and Correlation with Serum Glucose Levels in Rabbit"

_biosensors, 2021, doi:10.3390/bios11100387_

Round 1

Reviewer 1 Report

The authors introduce and thoroughly describe a novel miniaturized optical device that empowers the monitoring of glucose variations in the aqueous humor of rabbits' eyes. The manuscript is concise and clear and no major changes are needed. 

The device here introduced for the first time may cause a paradigm change in the management of diabetes. It may really push towards the development of devices that account for glucose variations in a non-invasive way.

The authors are however required to provide additional details about the experimental setup. They need to clearly describe the type of laser diode, photodiodes, and beam splitter used to build the optical device. The future readers need to be put in the conditions of replicating and testing the device.

Author Response

Response:

We deeply appreciate your valuable comments and suggestions. We have addressed concerns that you have raised in the following responses.

The optical polarized light sources are laser diode modules at wavelengths of 1650 nm emitting at 5mW. After the light was emitted from the light sources, it passed through a beam-splitter and was divided into two light beams, which reached the testing eyes and the photodiodes (for optical feedback control) respectively. The light reflected back from the testing eye was also divided into two light beams by a beam-splitter and reached the two photodiodes that one photodiode for detecting polarization of the light and the other photodiode for detecting absorption of the light. We have expanded the description of the details about the experimental setup in the manuscript. \

Action:

On page 2, we have revised the paragraph to “The optical polarized light sources are laser diode modules at wavelengths of 1650 nm emitting at 5mW. After the light was emitted from the light sources, it passed through the beam-splitter, equipped with a focusing function, and was divided into two light beams, which reached the testing eyes and the photodiodes (for optical feedback control) respectively. The light reflected back from the anterior chamber of the testing eye was also divided into two light beams by the beam-splitter and reached the two photodiodes. The information of reflected light to the two photodiode detectors was measured including the optical rotatory distribution (ORD) and absorption energy of the light.”

Reviewer 2 Report

Could you add results of in vitro measurements of influences of urea, ascorbate and lactate?  How to compensate influences of fluctuations of concentrations of urea and lactate? Can you add a meaning of  an ophthalmologist on security of this method with regard to damaging of eye?  How frequently such measurement might be done? 

Author Response

Could you add results of in vitro measurements of influences of urea, ascorbate and lactate?  How to compensate influences of fluctuations of concentrations of urea and lactate? Can you add a meaning of  an ophthalmologist on security of this method with regard to damaging of eye?  How frequently such measurement might be done?

We deeply appreciate your valuable comments and suggestions. We have addressed all concerns that you have raised in the following responses.

        Although ascorbate may be a source of interference, which may be restricted and excluded, and fluctuation of ascorbate is limited in the eye. On the other hand, the absorption coefficient and rotation angle of urea and lactate are much smaller than that of glucose, so the changing measuring results can be regarded as constants when glucose changes. We have added the results of in vitro measurements of influences of glucose, urea, ascorbate and lactate.

Contents

absorption coefficient

specific optical rotation

urea

0.058cm-1M-1 at 1650nm

lactate

0.208cm-1M-1 at 1650nm

 -13.5°

ascorbate

0.434cm-1M-1 at 1650nm

 +21°

glucose

0.478cm-1M-1 at 1650nm

 +52°

As for the security in the method in the aspect of ophthalmology, the measurement based on a non-contact technique, and reduced the risk of mechanical injury of ocular surface. As for the concern of light damage, the energy used in this method was low and the light was target on the iris of the eye. The pigments on the iris are able to absorb most of the light energy and prevent the light passing through the lens or reach the retina. From current safety information, the theoretical measuring frequency is not limited and can be used as need.

However, it may need further clinical trials for confirmation of the safety, efficiency, and protocol before the clinical use of the device, and the investigation is ongoing.